# Autophagy as a Therapeutic Target for Chronic Kidney Disease and the Roles of TGF-β1 in Autophagy and Kidney Fibrosis

**DOI:** 10.3390/cells12030412

**Published:** 2023-01-26

**Authors:** Miss Ruby, Cody C. Gifford, RamendraPati Pandey, V. Samuel Raj, Venkata S. Sabbisetti, Amrendra K. Ajay

**Affiliations:** 1Centre for Drug Design Discovery and Development (C4D), SRM University, Delhi-NCR, Rajiv Gandhi Education City, Sonepat 131029, Haryana, India; 2Renal Division, Department of Medicine, Brigham and Women’s Hospital, Boston, MA 02115, USA; 3Department of Medicine, Harvard Medical School, Boston, MA 02115, USA

**Keywords:** chronic kidney disease, autophagy, mTOR, TGF-β1, ATG, proximal tubular epithelial cells

## Abstract

Autophagy is a lysosomal protein degradation system that eliminates cytoplasmic components such as protein aggregates, damaged organelles, and even invading pathogens. Autophagy is an evolutionarily conserved homoeostatic strategy for cell survival in stressful conditions and has been linked to a variety of biological processes and disorders. It is vital for the homeostasis and survival of renal cells such as podocytes and tubular epithelial cells, as well as immune cells in the healthy kidney. Autophagy activation protects renal cells under stressed conditions, whereas autophagy deficiency increases the vulnerability of the kidney to injury, resulting in several aberrant processes that ultimately lead to renal failure. Renal fibrosis is a condition that, if chronic, will progress to end-stage kidney disease, which at this point is incurable. Chronic Kidney Disease (CKD) is linked to significant alterations in cell signaling such as the activation of the pleiotropic cytokine transforming growth factor-β1 (TGF-β1). While the expression of TGF-β1 can promote fibrogenesis, it can also activate autophagy, which suppresses renal tubulointerstitial fibrosis. Autophagy has a complex variety of impacts depending on the context, cell types, and pathological circumstances, and can be profibrotic or antifibrotic. Induction of autophagy in tubular cells, particularly in the proximal tubular epithelial cells (PTECs) protects cells against stresses such as proteinuria-induced apoptosis and ischemia-induced acute kidney injury (AKI), whereas the loss of autophagy in renal cells scores a significant increase in sensitivity to several renal diseases. In this review, we discuss new findings that emphasize the various functions of TGF-β1 in producing not just renal fibrosis but also the beneficial TGF-β1 signaling mechanisms in autophagy.

## 1. Introduction

Autophagy is a critical homeostatic mechanism that clears a variety of damaged or unwanted intracellular components to protect renal cells under stressed conditions. Autophagy deficiency, on the other hand, increases the vulnerability of the kidney to injury, resulting in reduced renal function, accumulation of damaged mitochondria, severe renal fibrosis, and early kidney failure [1,2]. The role of autophagy in fibrosis has been studied in a variety of ways. A complex communication pathway that detects changes in energy availability for activating or inhibiting autophagy has been identified in several studies [3]. Autophagy and the lysosomal degradation pathway aids in maintaining cellular homeostasis in the kidney cells. Any dysregulation in autophagy causes kidney diseases such as renal fibrosis [4]. Unbalanced autophagy can cause damage to podocytes, proximal tubular cells as well as glomerulosclerosis, as shown in Figure 1. After acute kidney injury, autophagy protects tubular cells from apoptosis and promotes cellular regeneration (as reviewed in [5]), although extended autophagy may lead to excessive digestion of vital cellular components and ultimately cell death [6]. Some of the most well-known proteins in the autophagic membrane are LC3, Beclin1, Atg 7, and Atg 12 [7].

Kidney fibrosis is identified by uncontrolled production and assembly of extracellular matrix (ECM) proteins in the interstitium of the kidney, causing organizational destruction, functional impairment, and eventually renal failure [8]. The buildup of scar tissue causes the expansion of the cortical-interstitial space, which is a predictor of chronic kidney disease [9]. Although the formation of a fibrotic framework after injury appears to aid tissue regeneration, it is eventually reabsorbed during interstitial remodeling following a mild injury, known as adaptive repair. CKD is characterized by maladaptive epithelial repair, comprising mitochondrial dysfunction, oxidative stress, abnormal autophagy, and tubular growth arrest, and apoptosis at the cellular level. Additionally, fibrotic matrix deposition proceeds unchecked in chronic injuries, disrupting organ design, function, and blood flow. This, in conjunction with nephron loss, causes kidney failure by impairing the body’s capacity to repair the functional tissues (reviewed in [8,9,10]). Numerous stimuli (TGF-β1, WNT, and connective tissue growth factor (CTGF)) and pathogenic factors (injury, diabetes, hypertension) can initiate the process, triggering wound healing and inflammatory signaling cascades that promote interstitial fibrosis [10].

In renal tissues, CKD is linked to significant alterations in cell signaling, such as the activation of TGF-β1, p53, and developmental genes such as Wnt and Notch [7,10]. Regardless of the etiology of kidney fibrogenesis, the TGF-β1/Smad3 signaling pathway is one of the key components that promotes the transcription of several profibrotic genes and drives fibroblast activation [11]. TGF-β activated kinase1 (TAK1), which activates the MAPK kinase (MKK)3-p38, are key upstream signaling molecules in TGF-β1-induced collagen I production [7]. According to a recent study, autophagy modulates the expression of TGF-β1 by proteolytic degradation and suppresses renal tubulointerstitial fibrosis in the unilateral ureteral obstruction (UUO) model. Increased TGF-β1 expression in LC3 deficient and Beclin1 (BECN1) heterozygous mice (autophagy-deficient) correlated with increased collagen I [11]. This study highlights a critical homeostatic negative feedback loop, where TGF-β1 promotes increased fibrogenesis while simultaneously upregulating autophagic processes that degrade mature TGF-β1 to terminate fibrotic signaling events.

WISP-1, a profibrotic protein, may promote renal fibrosis by activating autophagy in both obstructive nephropathy and TGF-β1-treated tubular epithelial cells (TECs) [12]. Autophagy inhibition makes TECs more vulnerable to TGF-β1 induced G1 cell cycle arrest and proliferative reduction, which leads to tubulointerstitial fibrosis (TIF) and nephron loss [13]. Nam, SA et al. found that autophagy in FOXD1-lineage stromal cells protects against renal TIF via downregulating TGF-β1/Smad4 levels through NLRP3 inflammasome signaling [14].

In this review, we focus on the role of endothelial and epithelial cells in maintaining the physiology and homeostasis in the kidney. We will discuss the role of activation and inhibition of autophagy in mitigating renal fibrosis via endothelial and epithelial cells. We will also elaborate on the process of kidney repair with the help or absence of autophagy-related genes (Atg), and how the repair is affected. Specifically, we will cover the role of TGF-β1 and its functions related to autophagy. Additionally, we will discuss the correlation between autophagy activation and lipid accumulation in renal fibrosis. There is also accumulating evidence that targeting autophagy can be used as a therapeutic strategy in other diseases and how that can be implicated in CKD to prevent maladaptive repair.

## 2. Kidney as a Homeostasis Organ

Maintenance of homeostasis is crucial for a normal healthily functioning kidney. The kidney is made up of many different cell types that are responsible for a variety of essential processes that keep the body in balance, including acid–base and osmoregulation, blood pressure control, hormone release, and nutritional absorption [3,15]. Reduced kidney function and renal damage are linked to acquired and hereditary problems in systemic iron homeostasis (as reviewed in [16]). Maintaining homeostasis in adult tissues is emerging as a physiological explanation of cell plasticity. Plasticity is studied extensively in the kidney because of the organ’s critical role in maintaining body volume, fluid osmolarity, acid–base balance, and the links between kidney illness and blood pressure regulation dysfunction [17]. CKD can induce fluid, electrolyte, and acid–base imbalances because of the kidney’s function in maintaining homeostatic stability [18].

The kidney plays a crucial role in maintaining ion homeostasis; it is responsible for managing the concentrations of ions such as calcium, phosphate, magnesium, and sodium levels in blood [19]. The bulk of sodium reabsorption occurs in the proximal tubule, but also occurs in the distal convoluted tubule. Sodium is transported across the apical membrane of the tubular epithelium by sodium-chloride symporters, which are activated by aldosterone and vasopressin. Kidneys maintain acid–base balance by regulating bicarbonate, hydrogen ion and ammonia content. The filtered bicarbonate is reabsorbed by the kidneys and the hydrogen ions and ammonia are secreted in the urine in the form of ammonium. Kidneys are also primarily responsible for controlling serum chloride levels. Most of the chloride filtered by the glomerulus is reabsorbed by both active and passive transport via both proximal and distal tubules (mainly proximal tubule) [19,20]. Below, we describe the role of constituent cells—podocytes, glomerular endothelial cells, and tubular epithelial cells—in maintaining healthy function and physiology of the kidney, and in turn homeostatic balance in the body.

### 2.1. Podocytes

Podocytes are cells attached to the glomerular capillaries at the glomerular basement membrane (GBM), generating an intercellular junction that makes a filtration barrier that helps maintain proper function [12]. Podocytes are crucial for blood filtration selectivity in the kidney, where they produce narrow slit diaphragms (SDs) by extending a web of foot processes (FPs) from their cell bodies that surround endothelial cells and interdigitate with those on neighboring podocytes [21]. Any damage to this location leads to glomerular disease. Podocytes that have been injured suffer effacement, which causes them to lose their structure and separate from the glomerular basement membrane, as shown in Figure 2. This results in a loss of the filtration barrier function. It is considered to be caused by a breakdown in actin filaments, which are intricate contractile structures that allow podocytes to rearrange in response to variations in filtration needs [22]. By activating chloride (Cl) conductance and raising calcium ions and Cyclic-AMP in podocytes, Angiotensin-II (ANG II) modulates the contractile state of their active foot processes [23]. Apoptosis has been suggested as a possible cause of podocyte loss [24], and due to the limited regenerative capabilities of the podocyte pool following injury, podocyte health is critical for maintaining healthy kidney function. Podocytes specialize in the maintenance of the capillary loop size, charge barrier to protein, help counteract intraglomerular pressure, synthesis, and maintenance of the glomerular basement membrane, and production of vascular endothelial growth factor (VEGF). It was illustrated that podocyte damage alters intra-capillary homeostasis, inducing thrombotic microangiopathy and signaling abnormalities by reducing VEGF levels [25]. EPB4.1L5, a FERM-domain protein that connects the actin cytoskeleton to cell membrane proteins is abundant in podocytes and holds a long evolutionary history. It has been discovered that remaining EPB4.1L5 from podocytes causes proteinuria, foot process effacement, and early death from kidney disease [26].

### 2.2. Glomerular Endothelial Cells

The endothelium of the kidney comes in a variety of shapes and sizes, each with its own structural and functional properties. The glandular endothelium, which is heavily fenestrated and coated in a thick glycocalyx contributes to the sieving qualities of the glomerular filtration barrier as well as the preservation of podocyte structure. The fenestrated microvascular endothelium in peritubular capillaries carries reabsorbed components [27]. In normal conditions, there is very little albumin in the GBM or overlying podocytes, revealing that the endothelial layer acts as a protein barrier. This study highlights the significance of the glomerular endothelial glycocalyx as a first-line barrier preventing albumin filtration [28].

Glomerular endothelial cell (GEC) injury contributes directly to podocytes and mesangial cell destruction and leads to glomerular sclerosis [29]. A damaged endothelium glycocalyx, mitochondria damage and oxidative stress, abnormal cell signaling, and endothelial-to-mesenchymal transition are all signs of glomerular endothelial cell dysfunction. Glandular endothelial cell dysfunction is sufficient to trigger podocyte damage, proteinuria, and mesangial cell activation. Furthermore, endothelial marker expression declines in association with mesenchymal marker expressions such as smooth muscle actin (α-SMA) and fibroblast-specific protein 1, correlating with increased ECM protein synthesis. This suggests that GECs can take on a mesenchymal phenotype and may contribute to glomerular fibrosis (as reviewed in [30]).

### 2.3. Proximal Tubule Epithelial Cells

The mitochondria-rich proximal tubule cell provides the energy required for this segment to reabsorb roughly 60% of the filtered water and electrolytes. The apical surface contains a brush border to accomplish this, which increases the surface area accessible for reabsorption. Distressed proximal tubules are a rich source of cytokines and growth factors such as epidermal growth factors (EGF), which stimulate healing in the acute stage but can be harmful in chronic injury [31]. When the tubule is severely damaged, some of these cells may become dedifferentiated over an extended length of time. These dedifferentiated cells produce bioactive chemicals with autocrine and paracrine activities, which have negative consequences and exacerbate fibrosis [32]. Renal tubular epithelial cells heal themselves after sub-lethal damage and activate an extensive network of pathways that involves cell cycle checkpoints, proliferation, and cell death. It has been revealed that a substantial increase in the fraction of proximal tubular epithelial cells arrested in the G2/M cell cycle phase is linked to the advancement of fibrosis in acute or sustained kidney injury using several techniques to create varying severities of acute kidney damage [33].

## 3. Mechanisms of Autophagy

Autophagy is an evolutionarily conserved cellular recycling mechanism in eukaryotes. Autophagy minimizes cell damage and supports survival in the case of an energy or nutritional shortage, as well as responding to numerous cytotoxic shocks. It is produced under many situations of cellular stress [34]. Autophagy is vital for cell survival and maintenance because it degrades cytoplasmic organelles, proteins, and macromolecules and recycles the breakdown products. Micro-autophagy, macro-autophagy, and chaperone-mediated autophagy are the three main kinds of autophagy in mammalian cells. Macro-autophagy relies on the production of autophagosomes, which are cytosolic double-membrane vesicles that sequester and deliver cargo to the lysosome. Specific unfolded proteins are usually transported through the lysosomal membrane via chaperone-mediated autophagy. Micro-autophagy entails the direct ingestion of cargo via lysosomal membrane budding [35].

Autophagy depends on autophagosome synthesis, maturation, and intracellular re-localization, which leads to their integration with lysosomes. Not unexpectedly, individual aspects of the autophagy metabolic network, including endocytosis and phagocytosis, govern vesicular transportation, typically as signaling components [36]. Autophagosome formation is a signal for the start of autophagy, and the initiation step is mostly reliant on the Unc-51-like autophagy activating kinase 1 (ULK1) complex. ULK1 is a protein that acts as a serine/threonine kinase. Post-translational changes to ULK1, including phosphorylation and ubiquitination, are required for autophagy induction. Mammalian Target of Rapamycin Complex 1(mTORC1) and AMPK control ULK1 kinase activity [37]. The mTOR complex recognizes the nutritional condition of cells to start or halt autophagy. In mammalian cells, the ULK1-Atg13-FIP200 complex initiates autophagy. With autophagy-related protein 13 and the Atg13 stabilizer autophagy-related protein 101, ULK1 forms a stable complex [38,39]. Many autophagy-related targets, including beclin1 and the vacuolar sorting 34 protein (VPS34), are likewise phosphorylated by ULK1. Dapper1 (Dpr1) is thought to be a major regulator of the Beclin1-Vps34-Atg14L complex, which promotes autophagy. By decreasing Vps34 activity, Rubicon prevents autophagosome maturation [40].

LC3 was first identified as the light chain of microtubule-associated protein 1A and 1B in the rat brain (LC3A and LC3B) among the mammalian homologs of Atg8 [41]. LC3 is inserted into the prolonging phagophore membrane, which accumulates targets for degradation. Finalization of the autophagosome is accompanied by recycling LC3-II/Atg8 by Atg4, followed by fusion of the autophagosome to the lysosome, resulting in proteolytic degradation of engulfed molecules by lysosomal proteases [42]. LAMP-2 and the small GTPase Rab7 are involved in autophagosome–lysosome fusion (Figure 3) [43].

## 4. Autophagy in the Healthy Kidney

Autophagy is vital for the homeostasis and survival of renal cells such as podocytes and tubular epithelial cells, as well as immune cells in a healthy kidney. When compared to other glomerular cells, differentiated post-mitotic podocytes had stronger basal autophagy. The increased levels of basal autophagy in podocytes relative to other glomerular cell types are thought to be a cytoprotective quality-control mechanism that helps them avoid cellular deterioration as depicted in Figure 2. As post-mitotic cells, podocytes are unable to dilute potentially hazardous cellular components into their daughter cells. For the elimination of these components, they require high baseline autophagy levels. Autophagy induction in tubular cells, particularly in the PTECs, protects cells against stress such as proteinuria-induced apoptosis and ischemia AKI. In contrast to podocytes, tubular cells, which are non-mitotic, have modest baseline autophagy function under normal physiological circumstances [44]. After careful examination by numerous in vivo mouse studies involving renal injury, autophagy is currently recognized as a protective mechanism, both for sustaining cell survival during injury and for preserving the nephron morphology during the event (reviewed in [45]).

## 5. Autophagy and Chronic Kidney Disease

### 5.1. Fibrosis in Chronic Kidney Disease

CKD is the terminal stage of many kidney disorders, characterized by a glomerular filtration rate of less than 60 mL per minute for more than three months, which is driven by chronic changes in kidney morphology and function that has severe consequences for the patient’s health. It contributes to poor medical prognoses and an extreme financial burden on healthcare systems worldwide [46]. Renal fibrosis is the most prevalent clinical outcome of CKD despite the underlying injury or illness, and many signaling networks act simultaneously to influence disease outcomes. Insufficient repair from acute kidney injury has been linked to the worsening of CKD in a growing number of clinical investigations. This is corroborated by the discovery that inadequate tubular regeneration is linked to prolonged tubulointerstitial inflammation, fibroblast activation and proliferation, and extensive extracellular matrix (ECM) protein accumulation. Recent research has also shown that tubule-specific damage is enough to cause fibrosis, making it a key connection between AKI and CKD [47].

Glomerular filtration rate (GFR) is a well-established measure of renal excretory function and albuminuria is a sign of renal barrier failure. GFR and albuminuria are applied to categorize CKD (Glomerular injury). Both have been discovered to be accurate predictive indicators of CKD outcomes in the long run [48]. Numerous studies have linked reduced nephron counts to a higher risk of developing chronic kidney disease [49], and several new hypotheses have been described further to distinguish the relationship between adaptive and maladaptive repair. Tubular cells stalled in a dedifferentiated condition with the continued synthesis of profibrotic factors because of repeated insults following localized tubular epithelial damage contributes to the microvascular loss. JNK activation caused by tubular cells arrested in G2/M leads to fibrotic gene expression [50]. AKI has been linked to changes in DNA methylation and histone modification which results in altered transcription of genes linked to kidney damage, including tumor necrosis factor (TNF). These long-term changes lead to the development of CKD [51]. Kidney injury molecule-1 (KIM-1) is a vital biomarker for renal damage. Tubules positive for KIM-1 correlated with increased macrophage infiltration and pre-fibrotic regions show elevated expression of α-SMA, a fibroblast activation marker [52]. According to Humphreys et al., chronic KIM-1 expression causes inflammation and tubular interstitial fibrosis [53]. Even in normal settings, changes outside the chronically injured kidney lead to a state of relative hypoxia, with fewer peritubular capillaries and higher collagen accumulation, resulting in enhanced gaps between vessels and tubular cells. Reduced number of glomeruli due to injury results in hyperfiltration and higher tubular oxygen uptake in their respective tubules, aggravating the oxygen demand-delivery mismatch [8].

TGF-β1 is a versatile cytokine that has been shown to control a wide range of cellular activities including growth, differentiation, cell death, and healing as well as its important functions in pathology, such as bone disorders, fibrosis, and cancer [54]. TGF-β1 plays a key role in the start and advancement of renal fibrosis; its effector Smad proteins (Smad2, Smad3, and Smad4) have diverse and even antagonistic roles in fibrosis regulation. Smad3 can connect directly to Smad-binding sites within gene promoters to boost transcription; however, neither Smad2 nor Smad4 have DNA binding domains and instead operate as Smad3-based gene transcription regulators [55]. The autophagic factors responsible for inducing and inhibiting kidney fibrosis are depicted in a flow diagram in Figure 4. TGF-β1 signaling has been involved in the pathogenesis of illnesses such as connective tissue disorders, and fibrosis, and it is now well known that TGF-β1 controls a range of essential processes in normal growth and development and physiology [56].

Resident fibroblasts produce an interstitial matrix, which helps to stabilize the tubular section of the nephron. A distinct population of myofibroblasts has been regarded as the major origin of ECM in fibrotic development [57]. The functional components of the kidney are maintained by mesenchymal structures. Mesangial cells help to maintain the glomerular tuft of capillaries, which is responsible for the filtration process [58]. Modifications such as podocyte dedifferentiation, podocyte-mediated endothelial dysfunction, and podocyte-induced epithelial-mesenchymal transition all contribute to the development of kidney fibrosis [59]. Furthermore, mesenchymal pericytes aid in vascular repair following damage; however, when parenchymal elements are irrevocably lost throughout the damage process, mesenchymal elements replace the area with an extracellular matrix, ultimately causing kidney fibrosis.

### 5.2. Autophagy in Chronic Kidney Disease

Podocytes, being terminally differentiated cells, use autophagy rather than cell division to minimize the intracellular buildup of defective DNA and other unwanted macromolecules. Podocytes have a high level of basal autophagy which plays a significant role in maintaining podocyte integrity. Autophagy is a crucial intracellular mechanism for the viability of renal cells. However, either high or defective autophagy may result in podocyte damage. By increasing the activity of systemic NADPH, ANG II has been discovered to increase reactive oxygen species (ROS) generation and oxidative stress in the renal system, resulting in aberrant podocyte autophagy. Autophagy improved survival when cells are treated with ANG II, suggesting that podocytes increasing autophagy might be a potential target treatment for ANG II-induced podocyte damage [60]. The mTOR pathway activates autophagy, which protects phagocytes against apoptosis, foot process effacement, and the development of CKD, and rat and human podocyte studies have suggested that autophagy can be controlled through mTOR signaling [5]. Additionally, endothelial cells have been demonstrated to be regulated by autophagy during the transition process [29].

The loss of autophagy in podocytes significantly increases the sensitivity to several renal injuries. Mice with Atg 5 or Atg 7 loss of function mutations exhibited histological and clinical hallmarks of human focal segmented glomerulosclerosis (FSGS). Podocyte dysfunction was also observed when Atg 5 or Atg 7 is deleted [61]. Proteinuria is associated with a rapid development of end stage renal disease in patients with renal impairments. PTECs are exposed to excess proteins on the luminal surface, which promotes tubular atrophy and fibrosis, contributing to the progression of CKD. Excessive albumin is toxic to PTECs, and defective autophagy contributes to albumin overload-mediated proximal tubular cell toxicity. Albumin exposure, which is common in proteinuric conditions, reduces autophagosome number and impairs autophagosome function through an mTOR-dependent mechanism [62]. Free fatty acids from the bloodstream and the glomerular filtrate are constantly taken up by PTECs and used for ATP generation via beta-oxidation. Regardless, there have not been enough investigations on lipid metabolism in the kidney. A recent discovery demonstrates that autophagy slowdown is a novel mechanism of lipo-toxicity in PTECs [63]. Pharmacological autophagic manipulations are a possible technique for lowering kidney lipo-toxicity. Simple autophagy activation, on one hand, may result in increased lysosomal stress and phospholipid buildup. Autophagy inhibition, on the other hand, can phenocopy autophagy-deficient animals, resulting in increased lipo-toxicity. As a result, the accumulation of pharmaceutical phospholipids that restore autophagic flux might be a unique therapeutic strategy [63].

Autophagy is also involved in the fight against inflammation, which is well-known to be linked to the development of CKD. Autophagy’s anti-inflammatory functions contribute to protection against the development of CKD. Interestingly, autophagy has the ability to modulate a variety of metabolic indices that are linked to renal damage, making it an attractive therapeutic target (reviewed in [64]).

Chronic and progressive kidney failure is caused by fibrosis, and the rate of Incidence is rising globally. Phosphoinositide 3-kinase/protein kinase B (PI3K/Akt) plays a critical role, acting as a signaling pathway to reduce autophagy. The mTOR (molecular receptor of rapamycin), which is highly evolutionarily conserved, is a well-known route that inhibits autophagy. In the obstructed kidneys of rats, autophagy increased followed by interstitial fibrosis, peaking after 3 days of UUO. Suppression of autophagy with 3-MA (3-methyladenine) resulted in reduced Akt/mTOR signaling, exacerbated tubular cell death, and interstitial fibrosis [65]. In other animal models, blunt suppression of mTOR by pharmacological or genetic methods may be advantageous, but due to the interconnectedness of these pathways, such action may have a detrimental impact on autophagy and raise cellular stress in the long run. Autophagy is a cell survival strategy, but it can also contribute to cell death as a result of chronic stress [66]. Autophagy can be profibrotic or antifibrotic depending on the contexts, based on the cell types and pathological circumstances. However, further study is needed to determine the best circumstances for initiating or suppressing autophagy to avoid chronic kidney disease progression.

To evaluate whether targeting autophagy is a realistic technique for the prevention or treatment of kidney illnesses, additional research is needed employing medications that are more selective for autophagy, as well as animal models with inducible autophagy modulation prior, during, and after kidney damage.

#### 5.2.1. TGF-β1 as an Inducer of Autophagy

Recent reports suggest a novel method through which TGF-β1 may provide cytoprotective benefits by inducing macro-autophagy [12,66]. TGF-β1 activated autophagy-related genes Atg5, Atg7, LC3, and Beclin 1 in kidney TECs, resulting in the formation of autophagosomes and the conversion of LC3 to the lipidated form, LC3-II. TGFβ-1 activates autophagy via the TAK1-MKK3-p38 signaling pathway, which increases collagen aggregation and breakdown of insoluble intracellular procollagen, suggesting a role for TGF-β1 in protection from fibrotic kidney disease [67]. Recent reports show that in vivo modulation of Beclin1 in mice protects from AKI-mediated kidney fibrosis development [68]. Livingston et al. reported that FGF2 secretion from injured PTECs PTECs led to the activation of fibroblasts and leads to kidney fibrosis. Fibronectin was deposited in the cells along with autophagy induction [69]. Furthermore, upregulation of WISP-1 in TECs promoted autophagy, as demonstrated by enhanced GFP-LC3 and LC3 and Beclin 1 expression in response to TGF-β1. Reports suggest that TGF-β1 induces autophagy via the TAK1-MKK3-p38 signaling axis, which protects renal mesangial cells from death during serum deprivation [12,70]. Additionally, (TAK1)-binding proteins 2 (TAB2) and TAB3, two of TAK1’s binding partners, have recently been shown to be endogenous autophagy inhibitors [71]. Thus, these reports confirm the role of TGF-β1 as an inducer of autophagy.

#### 5.2.2. Autophagy: A Regulator of TGF-β1

Numerous functions of TGF-β1 demand that its signaling be closely controlled at several levels. TGF-β1 activates Smad7, an inhibitory Smad that regulates TGF-β1 via a negative feedback loop [54]. Autophagy has recently been discovered to have a function in the regulation of the IL-1 family of cytokines. Like TGF-β1, IL-1β is a pro-inflammatory cytokine that is first generated as a pro form and regulates autophagy. IL-1β secretion by directing intracellular pro–IL-1β for destruction. New results point to autophagy as a cytoprotective mechanism that limits TGF-β1 secretion and suppresses the establishment of interstitial fibrosis in kidney injury by negatively regulating the generation of mature TGF-β1 proteins in RPTECs (renal proximal tubular epithelial cells) [54]. Increased TGF-β1 expression was also shown in UUO injured LC3 deficient and BECN1 heterozygous mice, correlated with increased collagen I [11], demonstrating that autophagy can, in fact, regulate expression of TGF-β1, uncovering yet another negative feedback loop, homeostatically programmed to limit TGF-β1 activity during renal injury progression. Thus, it has been shown that TGF-β1 plays a dual role in regulating autophagy, and there is a need to further investigate the role of TGF-β1 in autophagy to understand its true therapeutic potential.

#### 5.2.3. Other Regulators of Autophagy

Several signaling pathways influence autophagy. The mTOR pathway controls macro-autophagy. Ragulator, in conjunction with V-ATPase and Rag, may detect intracellular amino acid levels [72]. The LC3-interacting region (LIR) motif is regulated by post-translational modifications in other autophagy receptors, blocking or increasing the interaction with Atg8 family members. Post-translational modifications influence ATG8 proteins as well [73]. Even though mTORC1 activity is suppressed, autophagy initiation is suppressed, and cap-dependent mRNA translation is sustained during mitosis, according to recent research. A switch from mTORC1 to cyclin-dependent kinase 1 (CDK1)-mediated regulation is a significant contributor to this process [74]. Through its metabolite leucine, acetyl-coenzyme A inhibits autophagosome biogenesis (AcCoA). Acetyl-coenzyme A suppresses autophagy through increasing EP300-dependent acetylation of mTORC1 component raptor, resulting in mTORC1 activation. Interestingly, the major impacts on autophagy in leucine deficiency circumstances are driven by diminished raptor acetylation, which inhibits mTORC1, rather than altered acetylation of other autophagy regulators [75].

A second set of macro-autophagy activation mechanisms has been discovered that is not dependent on mTORC1. One of these mechanisms includes the activation of adenylate cyclase, which increases cAMP levels, allowing inositol 1,4,5-triphosphate (IP3) synthesis via phospholipase C activation. The JNK1/Beclin1/PI3KCIII axis is the third well-known mTOR-independent macro-autophagy regulatory pathway. JNK1-mediated autophagy activation is seen in response to hunger, apoptosis, or elevated nitric oxide levels in the cytosol [76].

#### 5.2.4. Autophagy Deficient Kidney

By digesting and reusing defective macromolecules and organelles, autophagy plays a critical role in metabolic reactions and cellular stability. For example, autophagy defects can promote endothelial-to-mesenchymal transition (EndMT). Autophagy is required for maintaining homeostasis in a variety of organs and cells and is particularly important in the kidney. Conditional autophagy-knockout mice (tissue-specific Atg5- or Atg7-KO mice) and autophagy-deficient cells have solidified the role of basal autophagy in cellular homeostasis in the kidney. In TECs, Atg5 has a detrimental effect on cell cycle progression, and autophagy suppression may enhance renal fibrosis via a G2/M cell cycle arrest-dependent mechanism [32]. In autophagy-deficient proximal tubular cells, protein aggregates and inclusion bodies have been reported [54]. Therefore, PTECs lacking autophagy is more vulnerable to a variety of stresses, and strategies to initiate autophagy when it is absent during fibrosis progression present an effective way to target and limit fibrogenesis and disease development.

## 6. Autophagy as a Therapeutic Target

mTOR inhibitors are one type of potential medicine that increases autophagy. Rapamycin (temsirolimus, everolimus, and ridaforolimus) and its derivatives (temsirolimus, everolimus, and ridaforolimus) are first-generation mTOR inhibitors that activate autophagy in yeast and mammalian cell lines [77]. In experimental models of diabetic kidney disease (DKD), the downregulation of autophagic pathways in podocytes has been discovered as a disease-promoting mechanism [78,79,80]. Sakai, S. et al. discovered that active autophagy in streptozotocin-treated diabetic mice neutralizes mitochondrial damage and fibrosis in the kidneys, but autophagic suppression in diabetic mice compromises kidney function even in the autophagy-competent condition [81]. In diabetic mice, ablation of Atg7 from the proximal tubules resulted in autophagy deficiency and worsened renal hypertrophy, tubular damage, inflammation, fibrosis, and albuminuria, indicating that autophagy plays a protective role in diabetic nephropathy [82].

Autophagy induction seems to be a promising treatment option for patients with various renal disorders. The mTOR inhibitors (rapamycin, sirolimus, and everolimus) are presently offered autophagy activators have shown to be effective in pre-clinical investigations employing a variety of animal models but have yet to show total therapeutic efficacy in patients with kidney disease. When developing therapeutic targets, it is vital to keep in mind that mTOR-dependent and mTOR-independent pathways exist for autophagy [83]. Pterostilbene inhibits TGF-β1-mediated NLRP3 inflammasome activation and epithelial-to-mesenchymal transition (EMT) in NRK-52E cells via inducing autophagy at the molecular level. To the best of our knowledge, this is the first study to look at the therapeutic benefits of pterostilbene in CKD via autophagy induction, which reduces NLRP3 inflammasome activation and EMT. Due to pterostilbene’s capacity to block NLRP3 activation and EMT through autophagy activation, the data indicate that pterostilbene might be a new preventative and therapeutic agent for CKD [84]. TGF-β1-induced kidney fibrosis is mediated by the NLRP3 inflammasomes, according to the findings. Zhang et al., showed that NLRP3 inflammasome aggregation and activation enhanced HMGB1 release via Gasdermin D, as well as mediating TGF-β-induced EMT. Isoliquiritigenin-mediated inhibition of the NLRP3 inflammasome lowers the secretion of HMGB1 and may help to mitigate fibrosis. TGF-β1 increased collagen I protein and mRNA levels in mouse kidney mesangial cells (MMCs). Bafilomycin A, an autophagy inhibitor that inhibits auto lysosomal degradation, increased collagen I protein levels without changing mRNA expression [85]. Collagen I colocalization with LC3 and LAMP1 was similarly elevated in this circumstance. Beclin1 genetic knockdown enhanced collagen I protein accumulation by TGF-β1, which was consistent with the pharmacological findings. Considerably, under normal conditions MMCs produced from heterozygous beclin1 knockout (beclin1+/−), mice showed higher quantities of collagen aggregates than wild-type cells. Moreover, TGF-β1 treatment further raised aggregated collagen I levels. These findings point to autophagy as a preventive mechanism against excess collagen buildup in the kidney [85].

Qi, YY. et al. investigated the relationship between autophagy induction and podocyte injury, including apoptosis, podocin derangement, albumin filtration, and wound healing to better understand the precise role of autophagy activation in a model of lupus nephritis. They discovered that many of the disease’s most important mediators—patient sera, IgG, and IFN-α—can induce autophagy in both murine and human podocytes. Furthermore, autophagy activation has a negative relationship with podocyte injury, which is responsible for proteinuria and the progression of glomerular diseases [86]. mTOR pathway-activated autophagy protects podocytes from apoptosis, foot process effacement, and the progression of chronic kidney disease [87,88]. Proteinuria and end-stage renal disease occurred in 3–5 weeks in podocyte-specific mTOR knockout mice [89]. Autophagosomes, microtubule-associated protein 1A/1B-light chain 3 (LC3), and damaged mitochondria accumulated in their podocytes [89]. This evidence suggests that autophagy in podocytes in both rat and human models is regulated by the mTOR pathway. The physiological level of mTOR activity inhibits autophagy and keeps autophagosomes at a low level in podocytes to remove damaged organelles, excess lipids, and long-lived or misfolded proteins [5]. Inhibiting autophagy by inhibiting mTOR using its pharmacological inhibitors also impairs podocyte function [90]. By 9 weeks, mice with podocyte-specific deletion of Vps34 had developed early proteinuria, progressive glomerulosclerosis, and renal failure. These knockout mice’s podocytes displayed a phenotype of impaired autophagic flux with an accumulation of enlarged vacuoles, indicating that Vps34 participates in maintaining autophagic flux in podocytes [91]. Likewise, podocyte-specific prorenin receptor-deficient (PRR^−/−^) mice developed nephritic syndrome within 2–3 weeks of birth and died by the fourth week [92]. Electron microscopy revealed that the mice had progressive podocyte damage including foot process effacement and vacuolation as well as podocyte cell death [92]. Adriamycin induces autophagy in cultured podocytes in vitro and in mouse podocytes in experimental models [93]. Importantly, inducible ablation of Atg7 in podocytes aggravated podocyte injury, glomerulopathy, and proteinuria during Adriamycin treatment, further implicating a protective role for autophagy in podocytes. Mice with proximal tubule-specific knockouts of Atg5 or Atg7 suggest that a low but sufficient level of basal autophagy is required to maintain cellular homeostasis in PTECs under normal conditions, whereas a higher level of autophagy is required to deal with age-related stress [33]. Under various stress conditions, autophagy is remarkably activated in PTECs and plays a reno-protective role against tubular injury and cell death [94].

### Clinical Significance of Targeting Autophagy and TGF-β1 Signaling

Studies have reported that autophagy protects PTECs from acute kidney injury in mice [95]. In an early diabetes mouse model expression of PINK1 was significantly increased in the kidney cortex, which suggests that mitophagy is increased in early diabetes [96]. In contrast, studies found that the PINK1 protein expression in PTECs of mice with diabetic kidney disease was significantly decreased [97]. Studies investigating the activation of mTOR signaling find that short-term treatment of mTOR activation protects podocytes in diabetes mellitus, but long-term activation can lead to proteinuria and progressive glomerulosclerosis [89,98]. Autophagy is protective in AKI, but it also inhibits complete repair in the long term [99]. Thus, targeting autophagy should be considered carefully as it may increase or decrease kidney injury.

Based on the research findings, there is a need for clear evidence that targeting autophagy will improve kidney health. This is because of cell type and disease type-specific functions of autophagy in the protection from and progression of injury. Further research is needed to identify important regulators which will be specific for targeting to alleviate CKD and its progression.

TGF-β1 is a renowned cytokine that plays a key role in the progression of organ fibrosis including kidney fibrosis (reviewed in [100]). Thus, targeting TGF-β1 signaling is a promising avenue for the treatment of fibrotic disease. TGF-β1-induced Smad-dependent and Smad-independent signaling directly activate many genes, producing extracellular matrix (ECM) proteins that are critical for fibrosis induction [101]. TGF-β1 neutralization lowers renal damage induced by high TGF-β1 activity in mouse models of kidney disease [102]. Inhibition of TGF-β1 signaling by deletion of TGF-βRII protects mice from UUO-induced kidney fibrosis [103]. Furthermore, TGF-βRII deletion from TECs inhibits tubulointerstitial fibrosis by inhibiting necroptosis and inflammation in mice [104], and in vitro, genetic silencing or pharmacological inhibition of TGF-βR1 in HK2 cells prevents fibrotic signaling and ECM deposition [105]. TGF-β1/Smads play an important role in the progression of AKI to CKD, where it has been shown that inhibiting TGF-β1 signaling worsens acute injury but its inhibition in the chronic phase is protective against fibrosis [106,107]. PTECs dedifferentiate and proliferate to replace injured epithelial cells during the injury. When the damage is severe and irreversible, certain cells fail to re-differentiate. These cells become arrested in the G2/M cell cycle phase and release profibrotic factors including TGF-β1 that cause fibrosis development. Unfortunately, clinical trials utilizing monoclonal anti-TGF-β1 antibodies with renin-angiotensin inhibitors did not slow the progression of diabetic nephropathy [108].

TGF-β1 is a critical mediator for renal fibrosis; hence, blocking TGF-β1 signaling might be a promising CKD treatment approach. There are several methodologies for clinically developing anti-TGF-β1 therapies for CKD. Pirfenidone, a non-specific antifibrotic agent shown to reduce production of TGF-β1, has been demonstrated to improve eGFR in DN and FSGS trials [109]. Bispecific antibodies against TGF-β1 and fibronectin’s extra domain A isoform protects mice from kidney fibrosis [110]. Based on the current knowledge, it is critical to identify more specific downstream pathways to inhibit TGF-β1 signaling. Ongoing clinical trials for TGF-β1 show potential for therapy in other diseases including various cancers but has yet to be translated into fibrotic disease.

## 7. Conclusions

We have reviewed the role of renal cells in maintaining homeostatic balance during injury progression through autophagy in great depth in this overview. While the role of autophagy and autophagy-related genes in distinct kidney cells has been highlighted in this review, the consequences of autophagy are complex and may change depending on the stage of kidney disease. Here, autophagy in renal cells and several approaches for preventing renal fibrosis are reviewed. TGF-β1 signaling, which is involved in the progression and modulation of fibrosis as well as the shift from AKI to CKD is well investigated. Autophagy is likely to be stimulated during AKI and repressed during CKD, and strategies to target downstream signaling pathways, such as NLRP3 inflammasome inhibition by pterostilbene to modulate autophagy could serve as novel therapeutic strategies to prevent the maladaptive repair in damaged renal cells during injury. Additionally, mTOR inhibition by rapamycin could serve as an alternative treatment. Thus, autophagy-regulating strategies may have major potential implications for preventing the progression of CKD.

## Figures and Tables

**Figure 1 cells-12-00412-f001:**
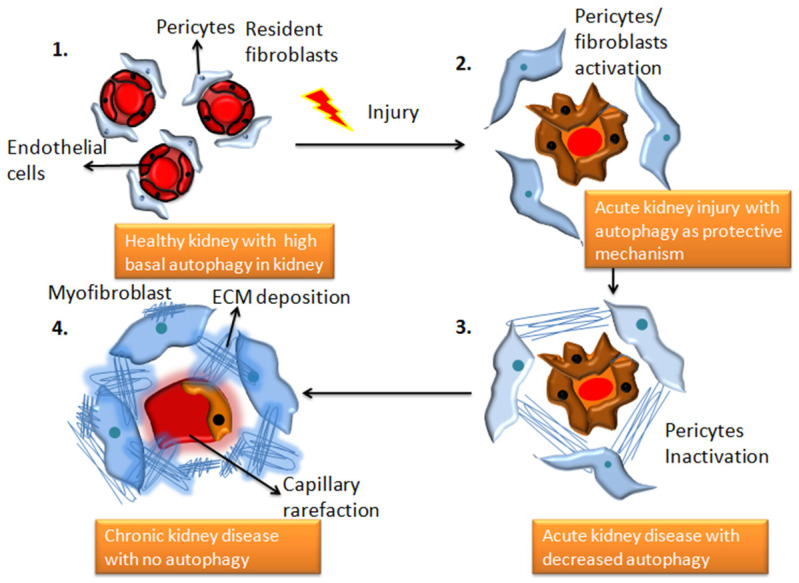
The transition from a healthy kidney to a chronically diseased kidney is associated with changes in autophagy patterns. 1. Demonstrates structurally healthy and physiologically efficient healthy kidney cells with high basal autophagy 2. Depicts the state of renal capillaries following damage caused by an underlying illness, toxins, drugs, or infection. In the extracellular areas, fibroblasts begin to release collagen and fibronectin proteins, and autophagy functions as a protective response, which can be reversible (adaptive repair). 3. AKI worsens to chronic kidney disease, and autophagic activities decline. This stage is characterized by morphological damage to the renal parenchyma and extracellular matrix deposition (maladaptive repair). 4. chronic kidney disease with significant fibrous protein deposition, and no evidence of autophagy.

**Figure 2 cells-12-00412-f002:**
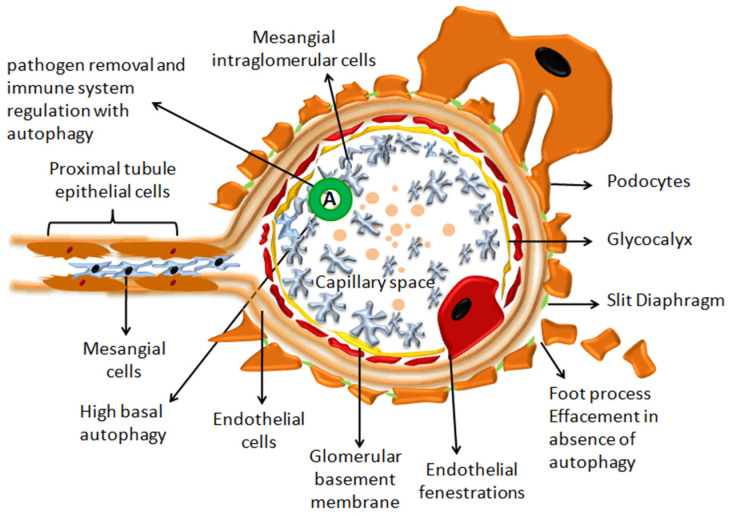
Podocytes are seen in the illustration surrounding capillaries in a glomerulus. Podocytes have higher basal autophagy levels than other glomerular cell types, which is a quality-control mechanism for cytoprotection. As illustrated above, foot effacement is seen in the absence of autophagy which leads to podocyte loss and is a morphologic hallmark of chronic kidney disease.

**Figure 3 cells-12-00412-f003:**
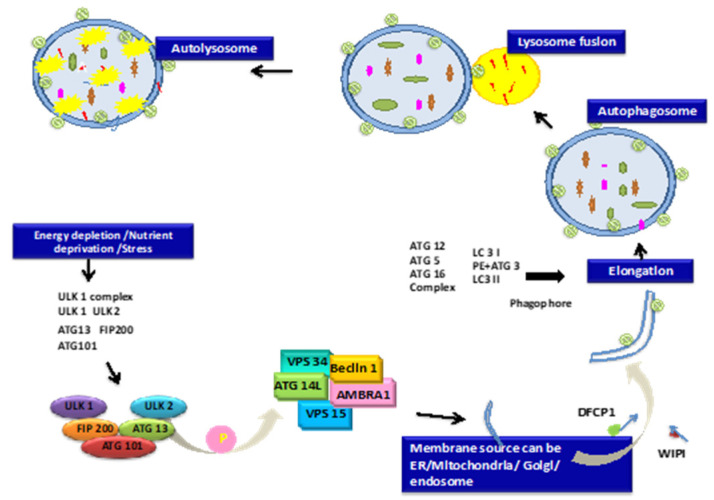
Molecular pathways involved in autophagy as explained thoroughly in the text above. Briefly, energy depletion leads to the activation of autophagy pathways by inducing LC3 signaling pathways. This leads to the formation of autophagosomes, which fuse into the lysosome forming autolysosomes.

**Figure 4 cells-12-00412-f004:**
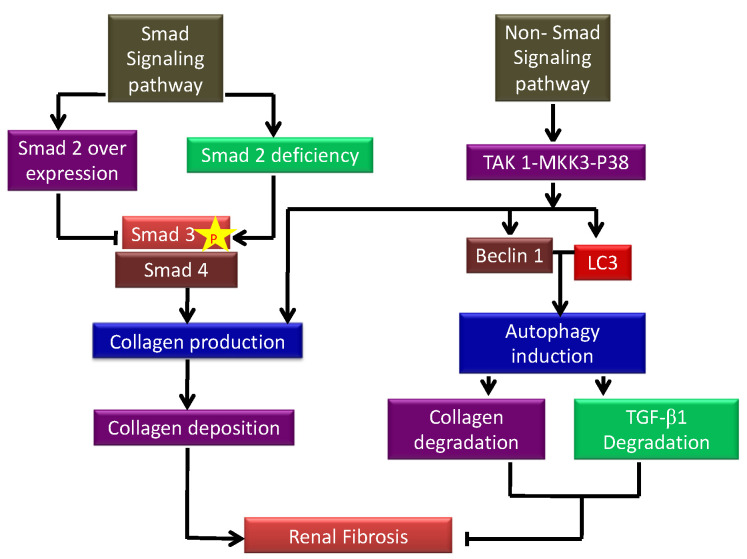
Schematic showing the autophagic factors responsible for inducing and inhibiting kidney fibrosis. p-Smad signaling increases collagen I production deposition leading to the development of renal fibrosis. On the other hand, non-Smad signaling inhibits fibrosis by inhibiting autophagy-mediated degradation of TGF-β1.

## Data Availability

Not applicable.

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
