# Peer review of "Autophagy as a Therapeutic Target for Chronic Kidney Disease and the Roles of TGF-β1 in Autophagy and Kidney Fibrosis"

_cells, 2023, doi:10.3390/cells12030412_

Round 1

Reviewer 1 Report

(1) Firstly, I suggested author should improve their English-writing. 

(2) References are required in some sentences. For example, lines 37-41

(3) Importantly, I think the structure of the article is disordered, the content is relatively empty, and some views are incorrect. For example, there are many functions for TGF-beta protein, including cell proliferation, growth, or autophagy,or other pathways.  They all play important roles in regulating fibrorsis.  However, authors did not address these points.

Author Response

Thank you for reviewing our manuscript.

Reviewer 1:

  1. Firstly, I suggested author should improve their English-writing. 

Response to the reviewer: Thank you for this positive comment to improve our manuscript. We have proofread the whole manuscript by a native English speaker and we have highlighted the changes in red color font texts.

  1. References are required in some sentences. For example, lines 37-41

Response to the reviewer: We have added the references to the whole manuscript.

  1. Importantly, I think the structure of the article is disordered, the content is relatively empty, and some views are incorrect. For example, there are many functions for TGF-beta protein, including cell proliferation, growth, or autophagy or other pathways.  They all play important roles in regulating fibrorsis.  However, authors did not address these points.

Response to the reviewer: We have corrected our views as per reviewer’s suggestions. We have added more details in all the sections to improve the quality of the review. We have restructured our review to have a better flow for readers.

Author Response

We thank the reviewer for evaluating our manuscript. We have highlighted the modifications in red font text in the manuscript.

  1. Abstract should be reviewed. First, you should give the definition of autophagy. After that, you should write about relation between kidney cells and autopahgy. Then, you could give information about kidney diseases and autophagy.

Response to the reviewer: Thank you for providing the positive comments to improve our review. We have added the definition of the autophagy followed by relation to kidney cells as suggested by the reviewer.

  1. Line 24, what did you mean with serum deprivation ?

Response to the reviewer: Yes, reviewer is correct, we had used serum deprivation. It is not in the revised abstract anymore, but we used Serum deprivation (SD) here to represent growth restricted conditions for showing cellular stress in vitro.

  1. Introduction should be reviewed. As in the abstract section, information about autophagy should be given first and then its relationship with chronic kidney disease should be mentioned.

Response to the reviewer: We have modified the introduction and abstract according to reviewer’s suggestions.

  1. “2. Kidney as a homeostasis organ” section. In my opinion, too much normal homeostasis information has been given in this chapter, beyond the subject of the review. This section needs to be simplified.

Response to the reviewer: We have made changes to simplify this section.

  1. The other sections should be written in the following order. “Autophagy in post CKD kidney repair” must be integrated into ”Autophagy in Chronic kidney disease”
  • Mechanism of autophagy
  • Autophagy in healthy kidney
  • Autophagy and Chronic kidney disease 1- Fibrosis in Chronic kidney disease.

2- Autophagy in Chronic kidney disease 2.1- Autophagy inducer TGF-β

2.2- Autophagy regulator TGF-β 2.3-Other regulators of autophagy 2.4- Autophagy deficient Kidney

  • Autophagy as therapeutic target.
  • Conclusions

Response to the reviewer: We have made changes to the order of the subsections as per reviewer’s suggestions.

Reviewer 3 Report

The manuscript covers quite nicely many aspects related to CKD and autophagy The manuscript is well written. There are some minor issues, which need to be considered by the authors.

1. The latest studies on how autophagy affects fibrosis should be discussed in detail. For example, in PMID 35491858, Tubular cells produce FGF2 via autophagy after acute kidney injury leading to fibroblast activation and renal fibrosis.

2. References should be checked. For example, Line 451. Please check Reference 77. In reference 77, the authors did not mention autophagy-related gene 7 (Atg7) from the proximal tubules

3. Some abbreviations should be only expanded the first time. for example Line 416  FSGS. (Focal segmental glomerular sclerosis) ; line 449, line 512 autophagy related gene 7 (ATG7)

4. line 529 “Autophagy-enhancing strategies may have major potential implications for preventing the progression of CKD.” Since autophagy play a diverse role in CKD, this conclusion should be carefully considered.  “Autophagy-regulating strategies” may be a better expression?

Author Response

We thank the reviewer for evaluating our manuscript. We have answered all the questions and comments raised by the reviewer and have highlighted them in a red colored font in the revised manuscript.

The manuscript covers quite nicely many aspects related to CKD and autophagy The manuscript is well written. There are some minor issues, which need to be considered by the authors.

  1. The latest studies on how autophagy affects fibrosis should be discussed in detail. For example, in PMID 35491858, Tubular cells produce FGF2 via autophagy after acute kidney injury leading to fibroblast activation and renal fibrosis.

Response to the reviewer: Thank you for suggesting the new concepts to be added to our review. We have already included this important article in our review and now have mentioned this new concept in our revised version.

  1. References should be checked. For example, Line 451. Please check Reference 77. In reference 77, the authors did not mention autophagy-related gene 7 (Atg7) from the proximal tubules

Response to the reviewer: We have included the right references as per reviewer’s suggestions.

  1. Some abbreviations should be only expanded the first time. for example Line 416  FSGS. (Focal segmental glomerular sclerosis) ; line 449, line 512 autophagy related gene 7 (ATG7)

Response to the reviewer: We have added these abbreviations in the revised version of the manuscript.

  1. line 529 “Autophagy-enhancing strategies may have major potential implications for preventing the progression of CKD.” Since autophagy play a diverse role in CKD, this conclusion should be carefully considered.  “Autophagy-regulating strategies” may be a better expression?

Response to the reviewer: We have made the changes in the conclusion as per the reviewer’s suggestion. 

Reviewer 4 Report

This review article by Dr Ruby et al, intends to summarize current knowledge regarding the role of autophagy in chronic kidney disease and the contribution of TGF-b signaling. Although potentially interesting, I have the following concerns regarding the manuscript in its current status:

1)    The vast majority of the listed references are other review articles, rather than original research articles. This is unpleasant for the readers, which want to verify or deepen the reported information.

2)    All the figure legends lack exhaustive/relevant explanations.  In detail: figure 1 shows different steps of kidney injury and autophagy activation which are not explained in the legend; figure 2 shows podocyte architecture but the legend refers to the higher basal autophagy levels in these cells, which is probably the main message the authors want to emphasize but it’s not shown in the figure; figure 3 lacks an exhaustive explanation of what shown.

3)    Some sentences should be corrected:

-In line 254, the cellular sensor of nutrient status is mTOR, not the ULK1-atg13-FIP200-Atg101 complex.

-in line 301 UUO acronym should be defined.

-in line 379, the reference is not correct.

-in line 392, the sentence is not correct, Ragulator is important for lysosomal recruitment of the RagGTPases, it’s not involved in Rheb axis.

-in line 439-440, the reported mTOR inhibitors are all ATP-competitors.

-in line 453, the reported inhibitors specifically target mTORC1.

Author Response

We thank the reviewer for evaluating our manuscript. We have answered all the questions and comments raised by the reviewer and have highlighted them in a red colored font in the revised manuscript.

This review article by Dr Ruby et al, intends to summarize current knowledge regarding the role of autophagy in chronic kidney disease and the contribution of TGF-b signaling. Although potentially interesting, I have the following concerns regarding the manuscript in its current status:

  1. The vast majority of the listed references are other review articles, rather than original research articles. This is unpleasant for the readers, which want to verify or deepen the reported information.

Response to the reviewer: We thank the reviewer for his positive comments on this review. We have now included the original research articles as the references.

  1. All the figure legends lack exhaustive/relevant explanations.  In detail: figure 1 shows different steps of kidney injury and autophagy activation which are not explained in the legend; figure 2 shows podocyte architecture but the legend refers to the higher basal autophagy levels in these cells, which is probably the main message the authors want to emphasize but it’s not shown in the figure; figure 3 lacks an exhaustive explanation of what shown.

Response to the reviewer: We have updated the figure legends to exactly explain the figures.

  1. Some sentences should be corrected:

-In line 254, the cellular sensor of nutrient status is mTOR, not the ULK1-atg13-FIP200-Atg101 complex.

-in line 301 UUO acronym should be defined.

-in line 379, the reference is not correct.

-in line 392, the sentence is not correct, Ragulator is important for lysosomal recruitment of the Rag GTPases, it’s not involved in Rheb axis.

-in line 439-440, the reported mTOR inhibitors are all ATP-competitors.

-in line 453, the reported inhibitors specifically target mTORC1

Response to the reviewer: We have corrected all these mistakes, and the last two have been omitted to make understanding easier for the readers.

Round 2

Reviewer 1 Report

I think the structure of manuscript is not good. In addition, authors did not give the significance of targeting TGF-beta for treating CDK. The clinical application of targeting TGF-beta or autophagy is required. 

Author Response

Response to the reviewer

We thank the reviewer for his suggestions.

Reviewer 1:

  1. The manuscript order is not good.

Response to the reviewer: We have added a new section on the clinical significance of targeting the autophagy and TGF-beta pathway.

  1. The authors did not give the significance of targeting TGF-beta for treating CDK. The clinical application of targeting TGF-beta or autophagy is required. 

Response to the reviewer: We have included a section for the significance of targeting autophagy and the TGF-beta signaling pathway with clinical application.

Reviewer 2 Report

Overall, this is a clear, concise, and well-written manuscript. The introduction is relevant and theory based. Sufficient information about the previous study findings is presented for readers to follow the present review rationale. The authors make a systematic contribution to the research literature in this area of investigation. Overall, this is a high quality manuscript that has implications for the theoretical basis of autophagy. I wish you success in your future studies.

Author Response

Thank you for accepting our manuscript.

Reviewer 4 Report

The authors corrected different mistakes and included more research articles among the references. Overall the quality of this review article is improved in the current version. 

Author Response

Thank you for accepting our manuscript.